# Study of the knowledge about gamification of degree in primary education students

**Alba Fiuza-Fernández**[1☯¤a], **Lucía Lomba-Portela**[1☯¤a]*, **Jorge Soto-Carballo**[2☯¤b], **Margarita Rosa Pino-Juste**[1☯¤a]

**1** Department of Didactics, School Organization and Research Methods, University of Vigo, Pontevedra, Spain, **2** Department of Socio-Educational Analysis and Intervention, University of Vigo, Pontevedra, Spain

☯ These authors contributed equally to this work.
¤a Current address: Department of Organization and Research Methods, Faculty of Education and Sports Sciences, Pontevedra, Spain
¤b Current address: Department of Socio-Educational Analysis and Intervention, Faculty of Education and Sports Sciences, Pontevedra, Spain
* lucialomba@uvigo.es

**Data Availability Statement:** All relevant data are available on openICPSR: https://www.openicpsr.org/openicpsr/project/163621/version/V1/view.

## Abstract

Gamification refers to the use of game mechanics in non-recreational environments, such as the school environment, in order to enhance motivation, concentration, effort, commitment and other positive values common to all games. Gamification allows us to establish clear objectives that are presented to be overcome. It also offers constant feedback, shows the progression of students, recognizes their effort and it guides them over the course of the teaching and learning process. The aim is to measure the knowledge of future teachers about gamification as a didactic resource. The sample is composed of 164 students of the Degree in Primary Education in Galicia (Spain). The scale obtains a reliability of 0.94 α. The index of Kaiser-Meyer-Olkin (KMO) provides a value of 0.932 and Bartlett's test of sphericity ($\chi^2$ = 2739,793; gl = 351, p < .000), ensuring that the factor analysis is right, and the model achieves a good fit. The students of the Degree in Primary Education have not heard of the term gamification, but still consider feasible its implementation in the school environment. The students feel that they don't know enough about this teaching resource, and they are afraid of not achieving the curricular objectives using it as they have no control over the content to be taught.

## Introduction

The technological advances we are facing in the society of knowledge are conditioning the model of social reality we know. These new realities are made of a whole set of worldwide changes that doubtlessly influence education, the economy, politics, all kind of social interaction processes, our spare time, and more, and they define what we today call the network society [1, 2].

In this context, we are witnessing a paradigm shift in education that exposes the ascension of new models of communication and information processing. Nowadays education and technology have a huge impact on the younger generation who consume data and they take for

**Funding:** The authors received no specific funding for this work.

**Competing interests:** The authors have declared that no competing interests exist.

granted that instantaneousness is a necessity and are connected all the time [3]. So clearly, one of the big challenges of today's education is to reformulate the process of learning, choosing the necessary means to meet the demands of the new students that are also digital native. These students show a lack of motivation and are not interested in classroom learning [4–6]. Prensky in his study on digital natives, confirms that "The students of today are no longer the kind of persons our education system was designed to teach" [6]. In many education forums, teachers are complaining that education is repressing the talents and capabilities of many students, suffocating their creativity and imagination, and creating a lack of interest in learning.

Digital natives, born from 1990 onwards, behave very differently from previous generations. They seem to be less worried about privacy and don't value face to face interactions, favouring texting over calling for socialization, studying or work. The key to change actual education lays in the digital devices and the closest games [7]. It is important these facts:

- They are students hard to surprise, because technology brought them closer to information and images in a way unthinkable in the last century; that's why they are so hard to motivate, make them feel curiosity, surprise them, etc.

- They have access to a huge amount of information, because everything is just a click away.

- As students, they get bored easily, and don't value personal efforts, because machines can do everything for them: difficult mathematical operations, complicated research, creative drawing, are a few examples.

- Material possessions are constantly in their minds, because they don't want to be without the latest videogame or phone model; they seek instant pleasure, enjoying the here and now. Everything is fast, so there is no lasting enjoyment.

- They always ask for one more chance, in the same way they have extra lives in their electronic games. They do not understand that in life, and in education, even though it's always possible to go back to school, some goals have their own "game over".

- They are individualist, but they like to work in groups to save time and effort, because their goal is not to learn but to pass the course.

Guided by these new needs of the students, we posit the use of gamification as an educational strategy in the classroom, because it increases motivation, engagement, experimentation, competition and group collaboration. Besides, gamification stimulates self-learning and the interest to keep learning or go deeper in certain topics [8–10].

The term gamification refers in particular to the translation of certain aspects of games to completely different situations, with the aim of improving motivation, active participation and engagement, of students in our case, in tasks that otherwise would be considered boring for them [11].

Authors like Zichermann y Cunningham [12] define the term gamification as "the process of game-thinking and game mechanics to engage users and solve problems". Others like Kapp [13] limit gamification to the use of mechanics, aesthetics and game thinking to attract people, motivate action, promote learning and solve problems. We are witnessing the gamification of everyday life. Children and teenagers use in their home a greater number of educational digital applications that in their schools.

Even though gamification [14] have been used successfully in marketing and human resources to improve productivity and develop the talent for innovation and creativity, education is just starting to think on what this game culture (videogames. . .) can add to the processes of teaching and learning [5] and what processes of behavior and procedure (autonomy,

engagement with the task, collaboration, experimentation and others) can be improve with this new reality. In education there have always been challenges to complete a task, but gamification means applying concepts and dynamics from game design in education. Furthermore, Batalla, Rimbau and Serradell [15] say the success of gamification may lie in its capacity to give credibility to its inclusion in the learning process.

In this emerging setting, it's important to adapt and develop digital strategies for the creation of learning resources [6], generate motivational synergies that connect with the interest of this new generation of students [16] and stimulate meaningful learning. Csikszentmihalyi proposes the idea of Flow, to show that humans have a mental state like a flow channel that keep them engaged with the task at hand and make them reach goals despite any negative emotions [17]. In this optimal state of motivation with a task, it's imperative that teachers have a clear understanding of the objectives, define accomplishable tasks, maintain constant feedback to correct any undesirable behavior, reach an equilibrium between the level of ability and the proposed challenge so that the activity doesn't turn up to be too easy or too complex to fulfil and that the activity itself is intrinsically gratifying.

From there emerges the idea of identifying game mechanics as a tool for learning using a competition model—where motivation [18], efficiency in the personal achievement [19] and the perception of competence [20] are more important every day—by using progress bars and ranking, and setting rewards—like prizes, levels and scores—as incentives.

The use of gamification rewards perseverance. The objectives are clear and are offered to the students as small challenges to overcome. Gamification offers a constant feedback, shows progress, rewards effort and guides the learning process. Using gamification, students will engage themselves with the game and in that way with their own learning.

Those reasons endorse the deployment, in a relatively short period of time, of gamification in the school environment, because it offers great possibilities for teaching and learning. But there are some models available, for example, *Gamification Model Canvas* [21] and *Business Model Canvas* [22], so the first question to ask is: do future teachers know of this strategy?

On the other hand, we must record the need of future teachers to teach skills related to new information and communication technologies because current generations have incorporated into their life the use of ICT and it is an advantage for working with these pupils.

Based on these premises, the main objective of this study is to describe the level of knowledge of gamification as a didactic resource and its use in university classrooms during the training of future teachers. Different research questions are posed: What is the level of knowledge that students have about the gamification as a resource, if they consider that it should be used and what are the difficulties they identify for their integration in the classroom.

## Method

As an approach into the reality of education, a transverse descriptive analytic study has been performed, because it tries to answer a theoretic problem and it is directed to describe reality [23]. This paperwork tries to find out the opinions and knowledge that future teachers of primary education have about gamification as an educational strategy.

Knowledge is measured through students' own self-perception, while opinions about the use of gamification as a resource and integration difficulties are vicarious experiences that have been seen but have not been personally experienced.

### Participants

The group studied is composed of the students of fourth and last year of the faculties of Science Education in the Degree in Primary Education, in the Autonomous Region of Galicia,

belonging to the University of Vigo (campuses of Ourense and Pontevedra), University of Santiago de Compostela (campuses of Lugo and Santiago) and to the University of A Coruña, with a total of 164 students. All of them have marked informed consent to participate in this study.

The average age is 23, with a minimum of 21 and a maximum of 48. As usual in education studies, there is a marked feminization, with 17.7% male and 82% female.

## Instrument

To answer the questions posed in this study, an ad hoc scale has been used to measure the knowledge of future teachers about gamification as a didactic resource. In order to know the opinion on the use of this resource in the school environment, three indicators have been identified: the possibility of application, the opportunities for use and its level of integration in the classroom. And, to know the difficulties for their integration in the classroom, they have asked directly about different situations that may lead to problems.

The instrument is composed of 4 questions (two closed-ended questions and two on a scale) and a Likert scale of 27 questions to be answered in a range from 1 to 5, with 1 being the minimal degree of agreement and 5 the maximum agreement with each one of the statements.

As reliability coefficient, Cronbach's alpha, was used, with a result of α 0.94, so the scale has an excellent internal consistency [24–26]. Table 1 shows the statistical indicators for reliability, by gender.

As a previous step to the selection of the extraction method to research the knowledge about gamification, maximum likelihood, two sampling adequacy indicators were calculated: a) Kaiser-Meyer-Olkin (KMO) index that offers a value of .932, considered very good and indicating that the correlations between pairs of items can be explained by the rest of the selected items, b) the Bartlett's test of sphericity ($\chi 2$ 2739.793; gl = 351, p < .000) showing that the items are not independents, guarantying that the factor analysis is suitable and the model obtains a good fit.

The interpretation of the structure of the factors gained after the analysis of the principal component identified two factors using varimax rotation. The two components explain the 51,15% of the variance of the date, with saturations from 0.50 to 0.82 showing the highest factor loadings in the items from 1 to 13 and the second factor containing the items from 14 to 27 with saturation varying between 0.60 and 0.68. To perform the analysis, items 2, 4, 7, 13 and 27 were inverted as they have a negative direction.

## Procedure

The strategy for the use of the instrument was to distribute the scale to the faculties using a form in Google Drive sent by email, stating that the participation was anonymous and voluntary, and that information was confidential. In the heading of the instrument, it is provided an explanation of what the term gamification means to clarify the concept to the students. In addition, an initial question was asked, after this concept was explained, the purposes of the investigation and the confidentiality of the data, which was stated as follows: After having

**Table 1. Statistical indicators for reliability.**

| Variables | | Cronbach's alpha |
|---|---|---|
| Gender | Male | 0.95 |
| | Female | 0.94 |
| Total | Scale | 0.94 |

considered the information they have given me, I declare that my decision is as follows: Yes, I give my consent; no, I do not give my consent. If the second option was checked, the survey is no longer completed and the survey is terminated.

The questionnaire was evaluated by the commission of the "Education, Sports and Health" doctoral program of the University of Vigo, who considered that given the nature of the study and that the participants were of legal age, it was not necessary to send the project to the committee of ethics since observational studies where personal data that can identify the interviewee are not requested or these are generally computed, only informed consent is necessary based on guidance from the Biomedical Research Law 14/2007.

Therefore, the study was conducted following the ethical standards laid down in the Declaration of Helsinki (Hong Kong revision, September 1989) and in accordance with the EEC Good Clinical Practice guidelines (document 111/3976/88, July 1990).

## Data analysis

To perform the statistical analysis, the program SPSS V.20.0 was used. The first step was to perform an analysis of the descriptive data of the variables in the study: frequencies and statistical summaries of the sample and each of the variables (averages and standard deviations) were obtained. Afterwards, the differences between means were studied, using Student's t-test as a parametric test for two independent samples, one-way ANOVA as a parametric test of K independent samples and Tukey's HSD for multiple comparison. For the scale variables, the Person correlation was calculated.

## Results

For this study, a test was carried out and students from different Galician Universities answered about their knowledge about the knowledge and use of gamification in the schools. Once the students know what it means, because they read information about it in the questionnaire before to fill in, they proceed to answer different questions about its use.

An initial global overview shows that the majority of the students (72.6%) have not even heard of the term gamification before fill in the questionnaire but, nonetheless, they think (85.4%) that it can possibly be applied in the school environment (Table 2) because of it characteristics as a didactic strategy.

After this short question, it is explained the meaning of gamification to release the possible use in classrooms.

Regarding the level of Integration of this tool in the classes of the curriculum in the university, we find a relatively low average ($\bar{X}$ = 2.31) but asking to pupils in different university classes (as a part of the questionnaire) how many opportunities they have had in the university of working in groups with the help of gamification, the average is very low too ($\bar{X}$ = 2.19) (Table 3).

**Table 2. Descriptive results of the level of knowledge and viability of application of gamification as an educational resource.**

| Variable | Item | Frequency | Percentage |
|---|---|---|---|
| *Level of knowledge* | Yes, I knew the term quite well | 33 | 20.1 |
| | Yes, I have heard the term and have a vague idea | 12 | 7.3 |
| | No, I have never heard it | 119 | 72.6 |
| *Viability of application* | Yes | 140 | 85.4 |
| | Depends on the age of the students and the subject | 14 | 8.5 |
| | No, too difficult | 10 | 6.1 |

**Table 3. Descriptive results regarding level of integration and its use in university classrooms.**

|  | *Level of integration* | *Opportunity of use* |
|---|---|---|
| *N* | 164 | 164 |
| *Average* | 2.31 | 2.19 |
| *Median* | 2.00 | 2.00 |
| *SD* | 1,111 | 1.149 |
| *Asymmetry* | .446 | .630 |
| *Kurtosis* | -.747 | -.607 |
| *Minimum* | 1 | 1 |
| *Maximum* | 5 | 5 |

The Symmetry and Kurtosis of the two variables are very similar. The symmetry values clearly show that the distribution has an asymmetrical tail skewed toward negative values, that is, the elements of the sample are on the whole skewed toward the lowest level. In the case of the Kurtosis measure, the distributions turn out to be negative. A negative Kurtosis value denotes a relatively flatter distribution, that is, Plattykurtic, meaning a lower concentration of data around the average.

Regarding the reasons that the students give for their reluctance to use gamification, we find the following data.

As can be seen, in Table 4, the highest percentages are located in the level 4 and 5, showing total agreement with the statement. In fact, almost all students (76.7%) think they have not enough knowledge to use this educational resource and state they fear that the use of gamification will prevent them from reaching the objectives established by the school curriculum (71.1%). In addition, they think that gamification is only good for games but not for learning (75.7%) and that they have no control over the educational content to be taught (71.9%).

Interestingly enough, despite an average age of 23, they think they are too old to gain a complete enough knowledge of this resource (57.9%) and lack information about the educational usefulness of gamification (58.8%).

The distribution of answers in Table 4 is much more uniform when asked about the usefulness of gamification or they time needed to use it.

Regarding their opinion about gamification, the average is 4.0781 and the standard deviation is .390, showing a very favourable attitude towards this educational resource (Table 5).

Based on these descriptive data, in Table 6, we have tried to determine if there are differences of attitude between genders and we didn't find them (Table 6). Neither there are significant correlations with age. But there are differences regarding the variable university.

**Table 4. Descriptive results on the difficulties of using gamification.**

| *ITEMS* | *1* | *2* | *3* | *4* | *5* |
|---|---|---|---|---|---|
| *Fear of not reaching the planned objectives of the curriculum that can be guaranteed using other methods* | 0.9 | 5.6 | 22.4 | 44.9 | 26.2 |
| *Believing that gamification is not for learning, only for playing* | 2.8 | 9.3 | 12.1 | 36.4 | 39.3 |
| *Thinking that gamification is a complex resource and methodologically hard* | 3.7 | 16.8 | 31.8 | 28 | 19.6 |
| *Because lack of control over what they want to teach and learn* | 2.8 | 6.5 | 18.7 | 41.1 | 30.8 |
| *Fear of not being able to control the resource* | 1.9 | 9.3 | 12.1 | 47.7 | 29 |
| *High time investment needed to have a full knowledge of the resource* | 3.7 | 15 | 20.6 | 32.7 | 28 |
| *Too old to learn new and complex things* | 5.6 | 20.6 | 5.9 | 24.3 | 33.6 |
| *Because it's not a resource very useful in primary education* | 11.2 | 22.4 | 27.1 | 18.7 | 20.6 |
| *Having poor or no information about the usefulness of gamification* | 4.7 | 8.4 | 28 | 22.4 | 36.4 |

**Table 5. Result of student's t-test for the contrast of independent averages.**

| FACTORS | | Average | F | Sig. | Tukey's HSD Sig.[a] |
|---------|--|---------|---|------|---------------------|
| Scale | U.Santiago de Compostela | 4.1656 | 5.287 | .006 | U.Coruña- U.Vigo = .005 |
| | U.Coruña | 4.2395 | | | |
| | U.Vigo | 3.9429 | | | |

[a] The difference of averages is significant at level .05.

**Table 6. ANOVA results for contrast of independent averages.**

| FACTORS | Gender | Average | DT | F | sig | t | Sig. |
|---------|--------|---------|-----|---|-----|---|------|
| Total Scale | Male | 3.9949 | .68025 | 1.641 | .202 | -.862 | .390 |
| | Female | 4.0960 | .54863 | | | | |

Clearly, the University of A Coruña shows a more favourable attitude towards gamification as a resource that the university of Vigo. But there are not differences based of the level of knowledge of the Teaching students have about gamification.

As a first result, it is evident the lack of information in the three universities about this resource and the need to learn about it. As they understand the meaning of gamification, university students are receptive to the idea of using it in the classroom: applications and advantages.

## Discussion and conclusions

In the current socio-educational context, the need to change is a reality. On one hand, Prensky supports that "It's very likely that the brains of our students have changed as a result of their upbringing. That can be true or not, but we can state with certainty that their patterns of thought have change" [6].

With this study we measured the knowledge of future teachers about gamification as a didactic resource. However, more than half of the participants were unaware of the educational resource but did see its application in schools feasible.

For that reason, the new generations of teachers must meet the demands of the new students, the digital native. It's a generation not only very well versed in games, but that expect that anything that catches their interest has a game-like component [7]. Therefore, we think that living in a society where game culture is so prevalent and important forces education to consider closely to the possibilities that a learning strategy base on games can bring to the teaching and learning processes. Under this dynamic, gamification appears as a valid solution and presumably applicable to any school subject or student age. Although in the present study the participants have shown a great lack of knowledge about the educational resource. In addition, more than 70% of the participants affirm that they do not have training to be able to apply it.

The usefulness of gamification in education have been proven in different context. Nowadays, its implementation in the classrooms is still increasing [27–29]. Until now, some teachers apply gamification in they own classes, for instance, discarding traditional grades in favour of experience points [30].

The future teachers show fear of using an undiscovered technique. There is no information about the use of gamification in universities' classes that allows students to train and acquire a work technique. It exists a predisposition to learn but there aren't facilities inside the learning community.

There is a need of change inside the schools that includes the methodology of teachers. The universities or other courses should provide information of the application of this technique. This helps to work agility and empowers teachers to create their own activities adapting them to the characteristics of their class.

In fact, it can be stated that gamification increases the level of motivation of the student, and that they get better results in practical tasks and global grading; but, nevertheless, they show poorer performance in written tests and their participation in class activities is reduced [31].

The university students, the future teachers, feel they don't know enough about this educational resource and are afraid of not reaching the objectives of the curriculum if they use it because they can't control the educational content to teach and lack information about the teaching uses of gamification. Likewise, they feel insecure about the usefulness of the resource, or the time need it to use it in the classroom. However, this study shows that the term is welcomed into the future teachers although they don't have enough information. The universities should reformulate the curriculum to adapt the contents to this new social reality, increasing the ICT classes, practical classes, etc.

It is imperative to dispel the fear of innovation to produce a permanent renovation of the educational approaches that helps to create an education mainly centred in the expectations and needs of the students, an education better able to care for the diversity of contexts and the way the students learn, and closer to the realities of their everyday lives [32, 33].

Many of these opinions can be explained, as the study shows, by the low level of integration and the scant opportunities for teamwork using gamification in the different subjects taught in the Teaching curriculum. It's necessary that the initial training of the teachers ties with the new demands coming from society. Is this regard, Gutiérrez-Cabello, Losada and Correa clearly state "the fact that information technologies are not properly integrated in the initial training of teachers extremely hinders their inclusion in compulsory education, because the teachers, like the children, learn better from what they can see in use (in Teaching education) than what they are told in a decontextualized first year single class" [34].

So, the highly important to update the technological resources use in the different classes taught in Teaching. Right now, according to the latest Horizon report [35], gamification is receiving a greater and greater acceptance among researchers and teachers, because they can see that games encourage productivity and creative research in the students. For that reason, they expect that in two or three years gamification will be widely used in education because university education must adapt to the technological and social context where its students live and the classroom as space devoted to education and learning cannot be isolated from the outside world. Actually, in the near future (two or three years) gamification is expected to be a trend in virtual campuses [35].

Among the limitations of the study, we point out the need to expand the sampling to other universities and career paths to confirm these results. Professors should be taken into account this study about the usefulness of gamification in the classroom, because it is reflected the advantages throughout the text. Courses should be imparted from universities and schools to facilitate the adaptation of games to their students.

## Author Contributions

**Conceptualization:** Lucía Lomba-Portela, Jorge Soto-Carballo, Margarita Rosa Pino-Juste.

**Data curation:** Lucía Lomba-Portela, Jorge Soto-Carballo, Margarita Rosa Pino-Juste.

**Formal analysis:** Lucía Lomba-Portela, Jorge Soto-Carballo, Margarita Rosa Pino-Juste.

**Funding acquisition:** Jorge Soto-Carballo, Margarita Rosa Pino-Juste.

**Investigation:** Alba Fiuza-Fernández, Jorge Soto-Carballo, Margarita Rosa Pino-Juste.

**Methodology:** Lucía Lomba-Portela, Jorge Soto-Carballo, Margarita Rosa Pino-Juste.

**Project administration:** Lucía Lomba-Portela.

**Resources:** Alba Fiuza-Fernández, Margarita Rosa Pino-Juste.

**Software:** Lucía Lomba-Portela.

**Supervision:** Lucía Lomba-Portela, Jorge Soto-Carballo, Margarita Rosa Pino-Juste.

**Validation:** Jorge Soto-Carballo, Margarita Rosa Pino-Juste.

**Visualization:** Margarita Rosa Pino-Juste.

**Writing – original draft:** Lucía Lomba-Portela, Margarita Rosa Pino-Juste.

**Writing – review & editing:** Alba Fiuza-Fernández, Lucía Lomba-Portela, Margarita Rosa Pino-Juste.

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
