## [Decision Letter · Decision Letter 0]

12 Jul 2021

PONE-D-21-07412

Study of the knowledge about gamification of university students

PLOS ONE

Dear Dr. Lomba,

Thank you for submitting your manuscript to PLOS ONE. After careful consideration, we feel that it has merit but does not fully meet PLOS ONE’s publication criteria as it currently stands. Therefore, we invite you to submit a revised version of the manuscript that addresses the points raised during the review process.

We look forward to receiving your revised manuscript.

Kind regards,

José Gutiérrez-Pérez

Academic Editor

PLOS ONE

Journal Requirements:

3. Please include your tables as part of your main manuscript and remove the individual files. Please note that supplementary tables (should remain/ be uploaded) as separate "supporting information" files

'The funders had no role in study design, data collection and analysis, decision to

publish, or preparation of the manuscript.'

**Comments to the Author**

1. Is the manuscript technically sound, and do the data support the conclusions?

 Partly

2. Has the statistical analysis been performed appropriately and rigorously? 

Partly

3. Have the authors made all data underlying the findings in their manuscript fully available?

Yes

4. Is the manuscript presented in an intelligible fashion and written in standard English?

Patly

5. Review Comments to the Author

Reviewer #1: After reading the manuscript, I would like to make a series of considerations about the article presented. These should be understood as proposals for improvement that only aim to offer another point of view.

With reference to the abstract, the total number of participants should be included. This information is important for those researchers who read the paper. The abstract is the first text to be read and must offer relevant data that will favour the possible reading of the whole document.

I think it is a mistake to call the degree of the participating students as "Magisterio de la especialidad de Educación Primaria". At present, the degree has the denomination of Degree in Primary Education. The denomination to which the authors refer is closer to the previous curriculum whose degree was called "Diplomatura de Maestro especialista en Educación Primaria". The authors use the aforementioned denomination in several sections of the manuscript.

In the introduction, it is started that "These students show a lack of motivation and are not interested in classroom learning". This assessment seems to be a value judgment, since it is not supported by any author who has been able to point in this direction after empirical verification. I believe that students lacking motivation and not interested in classroom learning have occurred in almost all generations, but it is not a characteristic that defines students in general.

Another issue raised by the authors is in reference to gamification: "in tasks that otherwise would be considered boring [11]". Although this statement is extracted from González-Tardón, I do not fully agree with it. Gamification is not used because of the existence of "boring" tasks, but as a didactic strategy for teaching-learning (also for evaluation) that aims to promote student learning, without detriment to the already mentioned by the authors, as motivation, participation and commitment, to which the improvement of certain values and skills must be added. Other authors also include the acceptance of failure as something normal.

Again, I must disagree with the authors when it is stated that "…education is just starting to think on what this game culture…". It is true that it may be far from a democratized use in the classroom, but currently there is a group of teachers highly motivated by the use of other teaching-learning strategies and where gamification is not only tangentially in the daily work of teachers, but many of them make use of gamification as the backbone of all their students' learning. There are many groups of teachers (from different parts of the Spanish geography and also internationally) that have been united around this common interest. A stroll through any social network is enough to verify this union of teachers who share their experience and, in addition, share many of their own resources.

In reference to the research question, and its subsequent treatment in the manuscript, a series of doubts arise. The following lines attempt to make an evaluative synthesis of this aspect.

- There are three essential aspects that circumscribe the study problem: knowledge, use and difficulties.

- In the Methods section, it is reported that the aim is to ascertain "opinions and knowledge".

- In the instrument section, a scale to measure opinion is mentioned.

- The results focus on knowledge.

Opinion is a subjective idea about reality, which contrasts with knowledge obtained from verifiable and tangible information. Therefore, it is proposed to the authors to unify and clarify this issue in the manuscript.

I think that the "Methods" section should be called "Method", since only one method was actually used in the research.

In the subsection on participants, it is indicated that the total number of participants was 164. No mention is made of the study population, which a priori, I think to be the students of the Degree in Primary Education at the three Galician universities. In addition, information on the percentage of students who are single is included, a variable that is not used later and in this study seems to contribute nothing. On the other hand, the percentage of students per university and/or university campus is not specified. I believe that this information may be of more interesting than the aforementioned marital status.

There is a piece of information that is not correct. It is indicated that the group of participants is composed of fourth-year students between the ages of 18 and 48. It is impossible for a fourth year student to be 18 years old, as this student will necessarily be in the first year, but never in the fourth year. Therefore, the participants must be at least 21 years old, an age that corresponds to those students who have followed their normal academic education without repeating any year from the lower levels (primary education) to the studies that have given them access to university.

In the title of Table 1, which is found in the Instrument subsection, it is indicated that information is shown on the statistical indicators of reliability by sex and specialty. In contrast, the information in the table itself refers to gender and does not mention the specialty.

- First of all, it would be necessary to see whether the information on specialty is missing or whether this variable should be removed from the title itself. The authors themselves have mentioned at all times (although it has already been indicated that it was wrong) as a specialty of Primary Education. So it makes no sense to speak of specialty in Table 1, since all students must necessarily be of the same specialty. Another question is if the authors wanted to name a mention (which does exist in the Degree in Primary Education). If this were the case, the term specialty should be changed to mention and, in addition, the corresponding percentage data should be included.

- Another important aspect is the assumption that sex corresponds to gender. Although this aspect is not the focus of this paper, it is erroneous to equate both terms as referring to the same concept.

By way of illustration, sex (biological) is an aspect determined by chromosomes, genitalia, hormones and gonads; while gender is a sociocultural concept and is related to the feeling of belonging to a certain group. In certain cases, biological sex does not coincide with gender. In any case, the term should be unified, both in the title of the table and in the information included, using the term usedin the data collection instrument.

Regarding the instrument, it is indicated that two factors have been identified after the factor analysis. Authors are proposed to characterize each of the factors.

Focusing on the procedure, it is not clear how it was carried out. In addition, the fact that the data collection was online seems to clash with what is specified in the sample in which it is indicated that "informed consent was signed". I understand that it is possibly the expression of what happened, but really the students could only mark or accept that consent, unless they had to attach a signed document with such consent.

In the Results section there is again another question that is not entirely clear. This one is also related to previous sections.

- In the abstract, it is stated that the study participants "have not heard of the term gamification".

- In the procedure it is stated that "In the heading of the instrument, it is provided an explanation of what the term gamification means to clarify the concept to the students".

- In results, it is said that the students "answered about their knowledge about the use of gamification in the schools".

There are two different issues that are reflected in the ideas that precede these lines. One is to know about the term gamification and the other is to know about the use of gamification in the classroom. What was the aim of this research? The answer to that question would be the one on which all these assertions should revolve.

Also in results, it is stated "Once the students know what it means, they proceed to answer different questions about its use" and, in fact, the following paragraph shows the percentage on such aspect (here the term is being valued again). Instead, after table 2, it is stated "After this short question, it is explained the meaning of gamification to release the possible use in classrooms". This is not understood: were the students informed before starting to fill in the instrument, or were they informed after answering the two initial questions of level of knowledge and feasibility?

Similarly, if 72.6% had not heard of gamification, it seems that, no matter how much it was subsequently explained to them, those students could not have an opinion about feasibility, And much less, that it could be with an affirmation (85.4%). Researchers should conduct a deeper analysis on this information.

Further on, It is mentioned that "the reasons that the teachers give for their reluctance to use gamification". This aspect is not understood, since the instrument has been answered by students, so there is no information available from the teaching staff. It could be that they are referring to teachers in training, in this case this information should be included. This aspect should be clarified.

The assessment of the data in Table 4 is more than expected. If the percentage of students who do not know about gamification is high, it necessarily has to coincide with low knowledge for the use of this resource, its possibilities of use to achieve the objectives of the curriculum, the perception of low control of the contents and an erroneous opinion in thinking that gamification is just a game.

A mention is made of Table 5 but the data provided do not really coincide with the table included, since the latter focuses on presenting a test of contrast according to the gender variable (or sex, as appropriate). It seems that this table is more related to the information shown after it where this variable is mentioned, although Table 6 is mentioned instead, which seems to correspond to the university variable.

With respect to the information included in Table 6, the universities should be clearly indicated, since the U.S.C. denomination is not something common that many of the readers of the article may know, especially if they are not Spanish.

The idea "But, there are no differences based of the level of knowledge the Teaching students have about gamification" does not show any data pointing in that direction.

To end with the results section, it is stated that "it is evident the lack of information in the universities about this resource and the need to learn about it". We do not really have data from the universities; we only have data from university students, which is something much more concrete than speaking globally about the universities.

In the conclusions section, it is stated that "With this study our main objective was to investigate if gamification is a usual recourse". It has already been mentioned above that it was necessary to specify what we wanted to know with this work, whether the term gamification or the use of gamification in the classroom or both, but it should be specified more clearly.

We do not agree with the statement "The usefulness of gamification in education has not been proven yet, because it actual implementation in the classrooms is still emerging. Until now, very few attempts have been made to apply gamification techniques to education [26]". This statement is supported by the work of Lee and Hammer (2011). A decade later, it seems that the situation may have changed. In this regard, this reviewer has conducted a search in the WOS and the following data have been found:

- The first papers about gamification date back to 2011.

- The first paper that in addition to gamification focus on the university environment is published in 2012.

- If we add to focus on the area of Social Sciences, again it is the year 2012.

- And finally, filtering Spain as the context, we find that the first paper is published in 2013.

- Since 2013, according to these criteria there are a total of 211 papers published only in the WOS, so the real data could be much broader.

As can be seen, the reference in the document can only state that there have been few attempts to apply gamification techniques to education, since the above data shows that it is an incipient area of study at the time of publication. Currently, I do not believe that this statement still makes sense. Therefore, it would be advisable to look for more current work that can go in the same direction as the 2011 work.

Likewise, the statement "The future teachers show fear of using an undiscovered technique. There is no information about the use of gamification in universities' classes". There are studies on the use of gamification in university classrooms, not only at the international level, but also in the closest context of the researchers, Spain, as mentioned above.

On several occasions there are also references to "the future teachers". I think that it refers to the students of the degree in Primary Education as future teachers. In this case, it is not clear what is meant by the statement "the universities should reformulate the curriculum to adapt the contents to this new social reality". One question is the curricular adaptation to the current society, something that does not seem very appropriate either, considering that the current curricula have not been widely implemented over time. However, this aspect is not the subject of study in this paper. On the other hand, if an attempt is made to link gamification with the current curriculum, it is important to delimit this aspect. I personally believe that these are not two concepts that should be linked, since one issue is the curricular content of the curricula and another is the methodology used for the teaching-learning of this content and the acquisition of competencies. In this second case, it is not necessary to change the content, since the inclusion of gamification in the classroom is perfectly compatible with the current training of future teachers of Primary Education (and other university degrees). Something similar can be found with the proposal to increase the number of classes with ICT and practical classes. These two aspects are more related to the organization of teaching by the teaching staff, so they are just as viable with or without modification of the curricular content.

The statement "in the near future (two or three years) gamification is expected to be a trend in virtual campuses [31]" seems to be out of context. It is based on a paper published in 2015, so those two or three years have passed and, if true, we would already have data to corroborate it. Along with this, it is striking that they focus on virtual teaching since the title that has been taken as a reference in this work seems to be taught face-to-face, so we should contextualize this idea in the way of teaching of these future teachers.

I hope and wish that these evaluations can be useful to the authors for the improvement of the manuscript.

Reviewer #2: Thank you for the opportunity to review your paper titled “Study of the knowledge about gamification of university students”

Abstract

What was the objective of the work? The authors do not indicate it.

There are errors in citation throughout the text, such as, for example, Prensky, (year?).

Methods

Clearly establish the type of study carried out

What was the calculated sample size?

What was the response rate?

The sentence: “The average age is 23, with a minimum of 18 and a maximum of 48. As usual in education studies, there is a marked feminization, with 17.7% male and 82% female. 96.5% of them are single”. It is a result not a method.

What does this work contribute to the scientific field?

Reviewer #3: The main objective of the study is to describe the level of knowledge of gamification as a didactic resource and its use in university classrooms during the training of future teachers.

First of all, the topic is highly relevant as gamification is an emerging trend in the educational field. It is important to study whether future teachers know about it or not and if they are willing or not to implement it in their future careers.

However, some revisions are needed:

1) Literature review. Authors do not include some relevant studies in the field.

Authors claim that “there are not many models available” concerning gamification, clarification on what do the authors understand as “models” would be advisable given that this is a risky claim, as there are several recent works focused on gamification models such as:

de la Peña, D., Lizcano, D., & Martínez-Álvarez, I. (2021). Learning through play: Gamification model in university-level distance learning. Entertainment Computing, 39 doi:10.1016/j.entcom.2021.100430

Jamshidifarsani, H., Tamayo-Serrano, P., Garbaya, S., & Lim, T. (2021). A three-step model for the gamification of training and automaticity acquisition. Journal of Computer Assisted Learning, doi:10.1111/jcal.12539

Rutkauskiene D., Gudoniene D., Maskeliunas R., Blazauskas T. (2016) The Gamification Model for E-Learning Participants Engagement. In: Uskov V., Howlett R., Jain L. (eds) Smart Education and e-Learning 2016. Smart Innovation, Systems and Technologies, vol 59. Springer, Cham. https://doi.org/10.1007/978-3-319-39690-3_26

Floryan, M., Chow, P. I., Schueller, S. M., & Ritterband, L. M. (2020). The model of gamification principles for digital health interventions: Evaluation of validity and potential utility. Journal of Medical Internet Research, 22(6) doi:10.2196/16506

In addition, authors claim that “The usefulness of gamification in education have not been proven yet, because it actual implementation in the classrooms is still emerging.” This claim is completely inaccurate. There is extensive literature on the effects of gamification in education, both in Higher Education as well as in Primary and Secondary education, to name but a few:

Buckley, P., & Doyle, E. (2016). Gamification and student motivation. Interactive Learning Environments, 24(6), 1162-1175. https://doi.org/10.1080/10494820.2014.964263

Barata, G., Gama, S., Jorge, J., & Gonçalves, D. (2013). Improving participation and learning with gamification. In Proceedings of the First International Conference on Gameful Design, Research, and Applications - Gamification ’13 (pp. 10–17). Stratford, Ontario, Canada: ACM. https://doi.org/10.1145/2583008.2583010

da Rocha Seixas, L., Gomez, A. S., & de Melo Filjo, I. J. (2016). Effectiveness of gamification in the engagement of students. Computers in Human Behavior, 58, 48–63. https://doi.org/10.1016/j.chb.2015.11.021.

Diaz, S., Diaz, J., & Ahumada, D. (2018). A gamification approach to improve motivation on an initial programming course. In 2018 IEEE International Conference on Automation/XXIII Congress of the Chilean Association of Automatic Control (ICA-ACCA) (pp. 1–6). IEEE. https://doi.org/10.1109/ICAACCA.2018.8609701

Hamari, J., Koivisto, J., & Sarsa, H. (2014). Does gamification work? - A literature review of empirical studies on gamification. In Proceedings of the Annual Hawaii International Conference on System Sciences (pp. 3025–3034). IEEE Computer Society. https://doi.org/10.1109/HICSS.2014.377

Mekler, E. D., Brühlmann, F., Tuch, A. N., & Opwis, K. (2017). Towards understanding the effects of individual gamification elements on intrinsic motivation and performance. Computers in Human Behavior, 71, 525-534. https://doi.org/10.1016/J.CHB.2015.08.048

Rivera-Trigueros, I., & Sánchez-Pérez, M. M. (2020). Conquering the iron throne: Using Classcraft to foster students’ motivation in the EFL classroom. Teaching English with Technology, 20(2), 3–22.

2) Introduction/discussion section

The portrait given of digital native students is rather general and discouraging. Students are portrayed as selfish, materialistic, and whimsical. Aren’t there any positive features about Generation Z that can be mentioned? In addition, is this portrait based solely on the vision of the authors or is it taken from any research work, if so, it should be clarified in the text.

Furthermore, the authors claim that gamification is the most suitable tool for the new needs of these students, being those needs pretty negative as mentioned before. However, gamification relies on extrinsic motivation and the surprise factor will be lost over time, which could imply motivation and engagement to decrease. How can this be overcome for these students who are supposed to get bored easily, be hard to surprise, etc.? Perhaps it could be interesting to add some discussion on these topics.

3) Definition of the term gamification

The authors state in the Discussion section “That term refers to the use of videogames on the school as a part of the teaching-learning process what it is supposed to motivate and get an engage with the pupils.” Why are the authors referring to videogames for defining gamification? In fact, this definition of gamification does not correspond to those offered by the authors in the Introduction section, in which they defined gamification as follows “Authors like Zichermann y Cunningham [12] define the term gamification as “the process of game-thinking and game mechanics to engage users and solve problems”. Others like Kapp [13] limit gamification to the use of mechanics, aesthetics and game thinking to attract people, motivate action, promote learning and solve problems.” No specific mention to videogames is made in these definitions. Definitions of gamification should be revised so they are coherent

4) Methdology

Why do the authors use this reference “25) Sijtsma K. On the use, the misuse, and the very limited usefulness of Cronbach’s alpha. Psychometrika 2009;74(1):107- 120. doi: 10.1007/S11336-008-9101-0” to justify that their questionnaire is reliable?. Sijtsma argues that Cronbach’s alpha suffers from major problems

5) Results

It would be interesting to deepen these results “an initial global overview shows that the majority of the students (72.6%) have not even heard of the term gamification but, nonetheless, they think (85.4%) that it can possibly be applied in the school environment.” It is surprising that students think that gamification can be applied in schools even though they do not know what it is.

---

## [Author Response · Author response to Decision Letter 0]

13 Oct 2021

Dear reviewers, 

First we would like to thank the reviewers who have corrected the manuscript. We would really like to convey our sincere thanks for each, and every suggestion that they made. With them, we have been able to significantly enrich the article sent to the journal. We are very grateful. 

Here are all the suggestions made by each of the reviewers that we had included:

Reviewer 1: 

- We have changed the designation of our students to "Degree in Primary Education"

- We just changed the introduction phrases, as you had indicated in the corrections

- In reference to the research question, and its subsequent treatment in the manuscript, a series of doubts arise. The following lines attempt to make an evaluative synthesis of this aspect.

o There are three essential aspects that circumscribe the study problem: knowledge, use and difficulties. 

Yes, are them

o In the Methods section, it is reported that the aim is to ascertain "opinions and knowledge".

We added in method: Knowledge is measured through students' own self-perception, while opinions about the use of gamification as a resource and integration difficulties are vicarious experiences that have been seen, but have not been personally experienced.

o In the instrument section, a scale to measure opinion is mentioned.

We added to instrument: To answer the questions posed in this study, an ad hoc scale has been used to measure the knowledge of future teachers about gamification as a didactic resource. In order to know the opinion on the use of this resource in the school environment, three indicators have been identified: the possibility of application, the opportunities for use and its level of integration in the classroom. And, to know the difficulties for their integration in the classroom, they have asked directly about different situations that may lead to problems.

o The results focus on knowledge. The previous explanation allows to readers to understand the manuscript

- The information about single participant were excluded

- We changed the age from 18 to 21, it was a mistake

- We have unified the term gender

- We added the specialty to the title, as your suggestion

- The information about consent was modified because the term that we used was not the correct one

- The expression “the reasons that the teacher give for their reluctance…” the term teacher was a mistake and we changed for students, because we refereed to the “future teachers”

- As for the table 4 we analyzed their perception. If they considered that they have a low knowledge about gamification but with the information that they received is enough to know that it is an useful didactic strategy

- We changed the place of table 5 to 6 and vice versa

- We also added the U.S.C for U. Santiago de Compostela in the Table 6

- In the conclusions we added information from nowadays to complete aour affirmations

Reviewer 2: 

- We added the aim to the manuscript

- The method was completed

- The sample size was also completed

- The contribution of this manuscript it is included in discussion and conclusions

Reviewer 3: 

- The literature and also the discussion and conclusions were completed with more references

Finally, in reference to the other comments from the reviewer 2 and 3 we do not answer because the reviewer 1 just did it and we answered there. 

Thank you all, you comments were really helpful for us

---

## [Decision Letter · Decision Letter 1]

9 Dec 2021

PONE-D-21-07412R1Study of the knowledge about gamification of Degree in Primary Education studentsPLOS ONE

Dear Dr. Lomba,

Thank you for submitting your manuscript to PLOS ONE. After careful consideration, we feel that it has merit but does not fully meet PLOS ONE’s publication criteria as it currently stands. Therefore, we invite you to submit a revised version of the manuscript that addresses the points raised during the review process.

We look forward to receiving your revised manuscript.

Kind regards,

José Gutiérrez-Pérez

Academic Editor

PLOS ONE

Journal Requirements:

**Comments to the Author**

1. If the authors have adequately addressed your comments raised in a previous round of review and you feel that this manuscript is now acceptable for publication, you may indicate that here to bypass the “Comments to the Author” section, enter your conflict of interest statement in the “Confidential to Editor” section, and submit your "Accept" recommendation.

Reviewer #3: I would like to congratulate the authors for the work they have carried out concerning the revision of the paper. In my opinion, the paper is now strengthened, specially concerning the Methodology section. However, some of my previous revision comments are not addressed in the revised version on the paper, nor in the response to reviewers.

Comments about literature review:

Some works have been revised and added to the paper. However, some clarifications are still needed:

1) The authors changed the claim “there are not may models available” concerning gamification for “there are some models available”. Again, authors do not clarify what they understand by “models” and, given that there are some models available, it would be advisable to cite them and discuss them.

2) The authors state the following “The usefulness of gamification in education have been proven in different context, because its implementation in the classrooms is still increasing [26, 27,28]. Until now, very few attempts have been made to apply gamification techniques to education Some teachers apply gamification in they own classes, for instance, discarding traditional grades in favour of experience points [29]”. First, it is claimed that the implementation in the classroom is increasing but then, it is stated that very few attempts have been made to apply gamification techniques, which, is confusing as it was previously said that gamification is increasing. In addition, again, this claim is inaccurate as there are plenty of works concerning gamification in education. This paragraph should be revised for coherence and consistency.

Comments about the Introduction/discussion section

1) The authors have not clarified if the portrait given of digital native students (individualist, selfish, materialistic, etc.) is their own vision or not, as requested in previous revision. There is no clarification either on the revised version of the paper or in the response to reviewers.

2) “The shortage of studies on the topic can explain why the students of Degree in Primary Education”. In my humble opinion, this claim is too risky. There are in fact many works on gamification, if the authors refer to some specific aspect of gamification that has not been sufficiently addressed in previous research, they should precise it. If they refer to gamification in general, this claim is not accurate.

3) This comment has not been addressed, either on the revised version of the paper or in the response to reviewers.

Definition of the term gamification. The authors state in the Discussion section “That term refers to the use of videogames on the school as a part of the teaching-learning process what it is supposed to motivate and get an engage with the pupils.” Why are the authors referring to videogames for defining gamification? In fact, this definition of gamification does not correspond to those offered by the authors in the Introduction section, in which they defined gamification as follows “Authors like Zichermann and Cunningham [12] define the term gamification as “the process of game-thinking and game mechanics to engage users and solve problems”. Others like Kapp[13] limit gamification to the use of mechanics, aesthetics and game thinking to attract people, motivate action, promote learning and solve problems. “No specific mention to videogames is made in these definitions. Definitions of gamification should be revised so they are coherent

Comments about methodology

4) This comment has not been addressed, either on the revised version of the paper or in the response to reviewers.

Why do the authors use this reference “25) Sijtsma K. On the use, the misuse, and the very limited usefulness of Cronbach’s alpha. Psychometrika2009;74(1):107- 120. doi: 10.1007/S11336-008-9101-0” to justify that their questionnaire is reliable?. Sijtsma argues that Cronbach’s alpha suffers from major problems.

PLOS authors have the option to publish the peer review history of their article (what does this mean?). If published, this will include your full peer review and any attached files.

Reviewer #1: No

Reviewer #3: **Yes: **Irene Rivera-Trigueros

---

## [Author Response · Author response to Decision Letter 1]

10 Jan 2022

Reviewer #3: I would like to congratulate the authors for the work they have carried out concerning the revision of the paper. In my opinion, the paper is now strengthened, specially concerning the Methodology section. However, some of my previous revision comments are not addressed in the revised version on the paper, nor in the response to reviewers.

Comments about literature review:

Some works have been revised and added to the paper. However, some clarifications are still needed:

1) The authors changed the claim “there are not may models available” concerning gamification for “there are some models available”. Again, authors do not clarify what they understand by “models” and, given that there are some models available, it would be advisable to cite them and discuss them.

Thank you for this comment. We had included information about models. We decided to modified this information with two new references: 

(21) Hunicke R, Leblanc M, Zubek R, Mda: A formal approach to game design and game research. Proceedings of the AAAI Workshop on Challenges in Game AI, 2004;4:1-5.

(22) Osterwalder A, Pigneur Y. Business model generation: a handbook for visionaries, game changers, and challengers (Vol. 1). New Jersey: John Wiley & Sons, 2010.

2) The authors state the following “The usefulness of gamification in education have been proven in different context, because its implementation in the classrooms is still increasing [26, 27,28]. Until now, very few attempts have been made to apply gamification techniques to education Some teachers apply gamification in they own classes, for instance, discarding traditional grades in favour of experience points [29]”. First, it is claimed that the implementation in the classroom is increasing but then, it is stated that very few attempts have been made to apply gamification techniques, which, is confusing as it was previously said that gamification is increasing. In addition, again, this claim is inaccurate as there are plenty of works concerning gamification in education. This paragraph should be revised for coherence and consistency.

We also revised coherence and consistency. The final paragraph is:

The usefulness of gamification in education have been proven in different context, because. Nowadays, its implementation in the classrooms is still increasing [28, 29,30]. Until now, very few attempts have been made to apply gamification techniques to education some teachers apply gamification in they own classes, for instance, discarding traditional grades in favour of experience points [31]. 

Comments about the Introduction/discussion section

1) The authors have not clarified if the portrait given of digital native students (individualist, selfish, materialistic, etc.) is their own vision or not, as requested in previous revision. There is no clarification either on the revised version of the paper or in the response to reviewers.

Regarding of the portrait of digital native students as we indicate on the introduction is about the information of Prensky. 

2) “The shortage of studies on the topic can explain why the students of Degree in Primary Education”. In my humble opinion, this claim is too risky. There are in fact many works on gamification, if the authors refer to some specific aspect of gamification that has not been sufficiently addressed in previous research, they should precise it. If they refer to gamification in general, this claim is not accurate.

We refer about the formation that the students of Degree in Primary Education have about gamification. But we prefer now to take it out of the text to avoid confusions.

3) This comment has not been addressed, either on the revised version of the paper or in the response to reviewers.

Definition of the term gamification. The authors state in the Discussion section “That term refers to the use of videogames on the school as a part of the teaching-learning process what it is supposed to motivate and get an engage with the pupils.” Why are the authors referring to videogames for defining gamification? In fact, this definition of gamification does not correspond to those offered by the authors in the Introduction section, in which they defined gamification as follows “Authors like Zichermann and Cunningham [12] define the term gamification as “the process of game-thinking and game mechanics to engage users and solve problems”. Others like Kapp[13] limit gamification to the use of mechanics, aesthetics and game thinking to attract people, motivate action, promote learning and solve problems. “No specific mention to videogames is made in these definitions. Definitions of gamification should be revised so they are coherent

We wrote this affirmation “That term refers to the use of videogames on the school as a part of the teaching-learning process what it is supposed to motivate and get an engage with the pupils” in the first version of the article. But in the July´s correction we changed it because we realized that it was our mistake. Also, in the last version we did not include it. Thus, if you need any explanation, we are glad to received it. 

Comments about methodology

4) This comment has not been addressed, either on the revised version of the paper or in the response to reviewers.

Why do the authors use this reference “25) Sijtsma K. On the use, the misuse, and the very limited usefulness of Cronbach’s alpha. Psychometrika2009;74(1):107- 120. doi: 10.1007/S11336-008-9101-0” to justify that their questionnaire is reliable?. Sijtsma argues that Cronbach’s alpha suffers from major problems

Finally, thank you for this comment. We have decided to remove the reference because we had an interpretation error.

---

## [Editor Report · Decision Letter 2]

13 Jan 2022

Study of the knowledge about gamification of Degree in Primary Education students

PONE-D-21-07412R2

Dear Dr. Lomba,

We’re pleased to inform you that your manuscript has been judged scientifically suitable for publication and will be formally accepted for publication once it meets all outstanding technical requirements.

Kind regards,

José Gutiérrez-Pérez

Academic Editor

PLOS ONE

---

## [Editor Report · Acceptance letter]

16 Mar 2022

PONE-D-21-07412R2 

Study of the knowledge about gamification of Degree in Primary Education students 

Dear Dr. Lomba:

I'm pleased to inform you that your manuscript has been deemed suitable for publication in PLOS ONE. Congratulations! Your manuscript is now with our production department. 

Kind regards, 

on behalf of

Dr. José Gutiérrez-Pérez 

Academic Editor

PLOS ONE